# Mechanisms Underlying the Effects of Landscape Features of Urban Community Parks on Health-Related Feelings of Users

**DOI:** 10.3390/ijerph18157888

**Published:** 2021-07-26

**Authors:** Lin Zhang, Suyan Liu, Song Liu

**Affiliations:** College of Architecture and Urban Planning, Tongji University, Shanghai 200092, China; zhanglin@tongji.edu.cn (L.Z.); shupianniudiqi@icloud.com (S.L.)

**Keywords:** community park, park landscape, healthy feeling, restorative potential, healthy design

## Abstract

Urban community parks are closely related to the health of residents, and have a positive effect on residents’ perception of nature, alleviating anxiety, and promoting physical health. Many previous studies have examined the impact of community parks on the health of the population, but few studies have investigated the potential of specific landscape elements in community parks to restore physical health. We conducted psychological questionnaires with 440 users of community parks in Shanghai through on-site surveys. Based on the psychological questionnaire, a structural equation model of the relationship between the community park landscape environment and users’ feelings was established. The model indicated that the natural environment, activity environment, and rest environment in the community park had positive effects on the physical, mental, and social health of users. At the same time, we recruited 50 participants to conduct laboratory experiments examining physiological changes while participants viewed different types of scene photographs showing the same landscape element. By measuring physiological indicators, including skin conductivity and heart rate, we sought to identify the types of landscape elements that help relieve the stress of users. The results revealed that flower clusters and waterscapes in the natural environment landscape, plastic fitness trails and fitness equipment places in the sports area, landscape elements such as benches with backrests, Chinese style pavilions, and green corridors with plants in the rest space, played positive roles in alleviating feelings of pressure and promoting relaxation among community park users. Based on these findings, we propose specific design strategies to improve the landscape health of community parks.

## 1. Introduction

In 1989, the World Health Organization (WHO) proposed a new definition of health, describing health as a four-dimensional state involving the health of the body, mental health, positive social adaptation, and moral health. Improving elements of the natural environment at multiple levels promotes healthy lifestyles, enhancing the public health of the population [1]. As green spaces enabling residents in a community to carry out daily leisure activities, community parks are important for promoting the health of the national population in China [2].

### 1.1. Related Theoretical Support for Healthy Landscapes

Ulrich’s stress reduction theory (SRT) proposes that natural elements, such as plants and bodies of water, have the effect of calming emotions and relieving stress, aiding relaxation and recovery from stress in the natural environment [3].

According to the attention restoration theory (ART) proposed by Kaplan and Kaplan, long-term concentration can cause mental fatigue and distraction, and the natural environment can greatly alleviate this visual fatigue, leading to a feeling of relaxation. In addition, Kaplan and Kaplan proposed four characteristics of ART environments: being away, fascination, extent, and compatibility [4,5].

Appleton’s prospect-refuge theory (PRT) is based on the adaptive evolution perspective in geography, proposing that places that simultaneously take into account the dual functions of observation and shelter meet instinctive human needs. Shelter can help individuals avoid danger and conceal themselves, while lookouts enable them to observe their surroundings. Regarding healthy landscape design, this theory plays an important role in guiding designs that can make users in a space feel safe and reduce negative emotions [6].

Wilson proposed the biophilia hypothesis, in which humans have a close emotional connection to nature. According to this theory, because humans originate from nature and will eventually return to nature, they experience a natural sense of intimacy with nature. In addition, a high frequency of exposure to nature is proposed to alleviate various human diseases and improve the mental state. This theory laid the foundation for the subsequent development of the healthy landscape framework [7].

The American Horticultural Therapy Association (AHTA) developed a treatment method called horticultural therapy for treating patients suffering from physical or mental illnesses using plant cultivation or various gardening activities, ultimately promoting physical and mental health. The main treatment methods are psychotherapy, five sense therapy, occupational therapy, and play therapy [8].

### 1.2. Research on Healthy Landscapes

Empirical research on healthy landscapes in China and internationally has progressed through three main stages. The first stage was demonstrating the health effects of the natural landscape. Ulrich and colleagues compared participants’ mood and stress changes in the natural environment and the urban environment, concluding that the natural environment can improve participants’ positive emotions and alleviate stress [9,10,11]. Through studying natural landscapes, artificial landscapes and non-landscape environments, Tang reported that artificial landscapes can also have restorative effects, exhibiting greater restorative potential than non-landscape environments, and slightly less restorative potential than natural landscape environments [12]. Hansmann studied urban parks and forest landscapes, reporting no significant difference between the restorative potential of forest landscapes and that of urban parks, with urban parks exhibiting a greater effect on alleviating participants’ sense of pressure [13].

The second phase of healthy landscape research involved studies of the restorative potential of landscapes on a large scale, mainly based on deeper and more refined research informed by the first research phase, as well as the expansion and improvement of the field. Liu studied different types of natural landscapes, reporting that older people exhibited the greatest restorative potential in forest environments, followed by the lawn environment [5]. Miyazaki conducted research on different types of forest landscapes, revealing that the forest landscapes of Mosen had greater restorative potential than other types of forest landscapes [14]. An studied the physiological effects of coniferous forests and broad-leaved forests on participants, and found that high-density forest landscapes aroused fear in some participants [15]. Carrus studied the restorative potential of woodlands and protected areas and found that biodiversity in the natural environment had a positive effect on participants’ perception of restoration [16].

The third stage of healthy landscape research mainly focused on the restorative potential of small-scale landscape elements, involving even more in-depth examination based on the progress of research into large-scale landscape types. Kang studied the effects of paving, plant communities, and bodies of water on participants’ neurophysiological activity, revealing that plants had the greatest restorative potential, followed by bodies of water, and paving [17]. In a study of preferences for different types of landscape elements, Nordh reported that grass had the greatest restorative effects, followed by trees, people, water, and flowers [18]. Zhang studied the restorative effects of landscape elements in different types of park spaces, and found that water relieved participants’ feelings of pressure, whereas roads and gravel roads in the park increased the feeling of pressure [19].

The three phases of research described above progressed from basic research to more detailed investigations of the restorative effects of landscape elements on health. Based on previous research findings, the current study investigated different landscape patterns of the same type of landscape elements, including different colors of flowers, different combinations of patterns of waterscape spaces, different paving materials, and different areas of playgrounds, to clarify the impact of different types of landscape elements on participants’ health.

### 1.3. The Health-Promoting Effects of Landscape Elements

Community park landscape environments have positive effects on promoting physical health, mental health, and social adaptability of urban residents. Natural environmental elements, such as green vegetation, flowers and bodies of water in the park environment, have a positive effect on the maintenance and improvement of physical health. In environmental psychology, “arousal theory” is focused on the physiological and psychological aspects of the human body, examining the impact of the environment on people. People’s emotions are largely stimulated by the surrounding material environment, which directly increases the level of arousal. This change is manifested as a physiological increase in autonomic activity, including accelerated heartbeat, accelerated breathing, increased adrenaline secretion, and increased blood pressure [20]. People experience the natural environment through contact, watching, smelling and other behavioral activities, which exerts effects through green vision, plant volatiles, negative oxygen ions in the air, space microclimate and other mechanisms, inducing health benefits [21,22,23,24,25].

A large number of domestic and international studies have focused on the urban park environment and the health of urban residents, revealing that urban park environments can have an important effect on urban residents’ stress relief (SAT) and attention enhancement (ART) [3,4]. Green plants can relax the human nervous system, causing the body to be calm, comfortable and vigorous, which is conducive to promoting the mental health of the population [26]. Studies have shown that people who frequently use parks and parks and green spaces around their residences have lower levels of mental stress than residents who do not. In addition, a greater amount of time spent in parks has been associated with better mental state, and lower likelihood of suffering from mental illness [27]. Urban green space has also been reported to have a positive influence on emotional affect and arousal of users [28]. The natural environment can lead to people experiencing more positive emotions, including satisfaction, happiness, and peace, as well as alleviating negative emotions such as anxiety, fear, and anger [29].

In addition, parks play an important role in maintaining social relations. Studies have shown that park environments have the potential to maintain good social relationships among users, and the potential value is related to the usage rate of the park, which has a positive effect on neighborhood relationships [30]. Relationships between neighbors can be maintained through parks, and the establishment of repeated and short-term visual contact and dialogue contact is beneficial for strengthening relationships between neighbors [31,32]. For people with limited access to social communication conditions, such as those with poor health, retired older people, and children with limited mobility, parks play an even more important role in maintaining social relationships and building social adaptability of the population [33].

## 2. Materials and Methods

### 2.1. Overview

Based on previous research results, the current study investigated the mechanisms by which community park landscape elements influence users’ health-related feelings. Using a questionnaire survey method, users’ preferences regarding the perceived landscape health of a community park were obtained. In addition, a structural equation model of the internal relationship between the natural environment, the activity environment, the rest environment, and users’ health recovery in the community park was constructed. To examine human physiological data, we simultaneously recorded skin conductivity and heart rate while participants viewed landscape photographs of community parks. We then analyzed the impact of different types of landscape elements in community parks on the potential of human health recovery. We propose that the current findings can inform targeted improvements of community design strategies regarding the health promotion potential of park landscapes. The experimental design of the study is shown in Figure 1.

### 2.2. Experimental Setting

As a central city area, Jing’an District of Shanghai has the characteristics of high population density, urban function compound and perfect park and green space system. The parks are relatively comprehensive in planning and design, spatial layout, plant configuration, and equipment configuration, and the frequency of use of the park is high. Based on a comprehensive consideration of the type of community park, the area, the facilities, the surrounding population, the relationship and the geographical location of the surroundings of the park, we selected Zhongxing Greenland (Figure 2), Pengpu Park (Figure 3), Lingnan Park (Figure 4) and Sanquan Park (Figure 5) in Jing’an District as the research objects. These four community parks are characterized by a dense surrounding population and cramped public space, representing community parks in a high-density metropolis.

### 2.3. Classification of Landscape Elements in Community Parks

Grassland, shrubs, trees, flowers, waterscapes, rocks, mountain slopes, and animals in community parks have been reported to have high restorative potential [34]. The four dimensions of naturalness, perception, rest, and activity indicate that community parks have a positive effect on alleviating the feeling of mental pressure experienced in crowded places [35]. Based on existing research, the landscape elements are divided into three categories: natural environment, rest environment, and activity environment, with a total of 11 factors, as shown in Table 1.

### 2.4. Questionnaire Experiment

#### 2.4.1. Participants

As participants in the questionnaire experiment, we recruited individuals who use community parks. Because of differences in the areas of the four community parks and differences in the tourist capacity of the parks, we issued the number of questionnaires that corresponded to the proportional relationship based on the tourist capacity of the parks. Based on the calculation of tourism capacity, we sought to conduct a questionnaire survey of 440 users in four community parks. Because most community park users are older people, many of whom have visual impairments, the questionnaire was verbally relayed by the investigator to each user to facilitate the user’s understanding of the questionnaire. In total, 440 elderly people were consulted in the questionnaire survey, and 440 questionnaires were returned. The basic information of community park users is shown in Figure 6

#### 2.4.2. Health Perception Indicators

According to the function of community parks, bodily health, mental health, and positive social adaptability were examined. Mental health was investigated in terms of two aspects: attention restoration and stress reduction. Through the health promotion function of community parks, health perception indicators of community park users were constructed, as shown in Table 2.

#### 2.4.3. Procedure

The questionnaire focused on three components: basic information about community park users, evaluation of community park users’ health perception, and community park users’ evaluation of landscape elements. Regarding the health perception evaluation of community park users, the questionnaire surveyed the health status of users after their activities in the park. Specific questions are shown in Table 2. Regarding community park users’ evaluation of landscape elements, the questionnaire surveyed users’ perceptions and evaluations of different landscape elements. The specific types of landscape elements are shown in Table 1. Each item was rated on a five-level Likert scale: “strongly agree”, “agree”, “general”, “disagree”, and “strongly disagree”. The corresponding scores range from five points to one point.

### 2.5. Physiological Experiment

#### 2.5.1. Participants

We recruited 50 participants for the physiological experiment. All participants were healthy and did not have color blindness, and had not consumed any food containing tobacco, alcohol, or caffeine in the 12 h before the test. Before the experiment, all participants reported that they were in a good mental condition, did not perform any strenuous exercise 6 h before the test, and did not engage in behaviors that could harm their mental health, such as staying up late. The basic information of the participants in the physiological experiment is shown in Figure 7.

#### 2.5.2. Procedure

Previous studies have reported that indoor experiments using stimuli such as photographs, slides, videos, and virtual reality (VR) can be used to simulate outdoor environments in physiological experiments [36]. Using such methods, color slides or photographs can simulate the visual information received by participants in actual environments, and psychological and physiological responses can be elicited through the stimulation of the landscape elements. Based on the questionnaire survey, the current physiological experiment further explored the types and characteristics of specific landscape elements that affect the physical health recovery of users. The experimental stimuli included park greening rate (N1), plant color (N2), waterscape (N3) in the natural environment, fitness facility (E1), fitness track (E2), and activity space (E3) in the active environment, and seat (R1), pavilion and porch (R2) and green corridor (R3) in the rest environment. We presented three to five scene photographs for each type of community park landscape element, and each scene showed different characteristics of landscape elements.

The experiment used ErgoLAB HME Synchronization Technology, and testing was undertaken in a quiet indoor environment (Figure 8). In the preparation phase, the experimenter explained the experiment to the participants, and set up the physiological sensors. When fluctuations of the physiological data stabilized, experimental testing was begun (Figure 9) [37]. In the experimental test phase, participants sat quietly for 3 min and waited for physiological data fluctuations to stabilize, then viewed images presented on a computer screen. A total of 35 community park landscape scenes were used in the experiment, with each scene lasting for 10 s. To avoid interference between landscape scene photographs, black screen scenes were used in the transition period between scenes, presented for a duration of 5 s.

Heart rate response represents the emotional state of the subject, and changes in emotional mood, regardless of whether the mood change is positive or negative, lead to increases in heart rate. When the physiological experiment was finished, participants prioritized the landscape scenes, and the data were analyzed in combination with heart rate changes and scene preferences (Table 3).

#### 2.5.3. Physiological Indicators of Health Perception

With recent advancements in physiological measurement technology, many researchers have used medical equipment in empirical studies of health-related landscape features. Physiological measurement indicators of healthy landscapes include heart rate variability, blood pressure, electroencephalography, skin conductivity, salivary cortisol, skin temperature, muscle conductivity, blood oxygen saturation, and respiration measurement. In the current study, because participants viewed photographs of landscape elements for less than 5 min, common measurement methods could not be used in the experiment. Therefore, after pre-experimental analysis, skin conductivity and heart rate were selected as physiological test indicators, reflecting changes in the physiological characteristics of participants in a short period of time (Table 4).

## 3. Results and Discussion

### 3.1. Theoretical Model of the Relationship between Community Park Landscape Elements and Health-Related Feelings of Users

#### 3.1.1. Analysis of Health-Related Feelings of Users

In the evaluation of community park users’ health feelings (Figure 10), the average mental health score was 4.12, indicating that the community park environment had a positive effect on promoting people’s feelings of fulfilment, peace of mind, relaxation and pleasure, and relieving mental fatigue. The average physical health score was 4.11, indicating that users believed that the community park had good air quality, was conducive to exercise, and could alleviate chronic diseases. The average social adaptability score was 4.02, indicating that users believed that the community park had a friendly and comfortable atmosphere, producing a sense of belonging.

In the evaluation of the landscape elements of the community park (Figure 11), participants generally gave higher evaluation scores of the overall environment of the community park. The highest evaluation scores were given to the natural environment, followed by the activity environment. The greening rate in the natural environment, the fitness trails in the activity environment, and the seats in the rest environment were the most commonly evaluated landscape elements by users. The evaluation of all landscape elements was higher than the average (three points), indicating that the participants made positive evaluations of the landscape elements of the community park.

#### 3.1.2. Theoretical Model of Evaluation and User Perception of Landscape Elements in Community Parks

Importing 411 valid questionnaire data sets into SPSS19.0 statistical software for questionnaire reliability analysis (Table 5), the Cronbach’s α coefficient of the total scale was 0.916, indicating that the reliability of the questionnaire was very good. The Sig value of Bartlett’s test of sphericity was less than 0.01, indicating that the data passed Bartlett’s test; the KMO test result was 0.913, and the data were greater than 0.7, indicating that the questionnaire items met the standard (Table 6).

The data were imported into AMOS19.0, and the maximum likelihood estimation method was used to estimate the parameters of the model. According to the standardized model data analysis, the model’s varices analysis (Table 7) revealed that all of the data for this factor were positive and significant, indicating that the item indicators of the model met the standard criterion, and that there were no unacceptable items. The regression weights analysis of the model revealed that the *p*-values of the model assumptions were all less than 0.05, indicating that the structural path assumptions of the model were all valid. Overall, the results indicated that the untitled items of the model did not meet the standard criterion and needed to be deleted. The factor loading analysis of the model path revealed that all items had values greater than 0.6, indicating that the model factor loading met the standard criterion.

Figure 12 shows the estimation of the path coefficients of the structural equation model. The numerical values of the standardized path coefficient show the relationship between each measured variable and the latent variable, and the degree of influence of each measured variable on the latent variable. F1–F7 are latent variables, F1–F3 are exogenous latent variables, F4–F7 are endogenous latent variables, N1–N4, E1–E4, R1–R3, B1–B3, C1–C3, A1–A3, S1–S3 are all measurable variables, and e1–e27 are measurement residuals.

The results show that the three factors of community park activity environment, natural environment and rest environment exhibited positive effects on the bodily health, positive social adaptation, and mental health of users. The activity environment had the most importance and strongest effects on the bodily health of the user, with a standardized path coefficient of 0.54; the rest environment had the most importance and strongest effects on positive social adaptation of users, attention restoration and stress reduction, with standardized path coefficients of 0.55, 0.42, and 0.55, respectively.

### 3.2. Analysis of the Influence of Community Park Landscape Elements on Users’ Physiological Indicators

Based on the ErgoLAB software system, the 3500 pieces of data collected in the physiological Measurement indicators experiment were filtered and noise-reduced, and the variance analysis method (Liang, 2019) was used to determine whether the average physiological measurement indicators of different landscape types were statistically significant. Because of the large differences in individual physiological signals, it is necessary to remove the differences of physiological signals at a basic level to reduce the experimental error caused by individual physiological differences. Therefore, in the current study, the individual physiological response rate (R) of participants under different types of landscape stimuli was used as the research index of the stress relief test:
R = (X_experimental group_ − X_baseline value_)/X_baseline value_ * 100%
R: physiological level response rate.X_experimental group_: participants’ physiological signal data under different landscape types.X_baseline value_: physiological data of participants in a calm state.

The results revealed that the F value was greater than one, indicating that the difference between the means of each group was statistically significant. The landscape elements were associated with significant differences in participants’ skin conductivity, average heart rate, maximum heart rate, minimum heart rate (*p* < 0.05), indicating that the data of each group can be compared. Therefore, by measuring participants’ physiological data, we were able to examine the pressure-relieving effects induced by viewing photographs of different landscape elements.

#### 3.2.1. Analysis of Physiological Indicators under Different Park Greening Rate Scenarios

The greening rate of the community park (N1) was shown in three scene photographs: a large area of lawn (N1.1), a general sparse forest space (N1.2), and a dense forest space with low openness and rich plant layers (N1.3). We examined the impact of park greening rate and its openness on the emotional recovery of participants. The specific scenarios are shown in Table 8.

When participants watched N1.3, the physiological response rate of skin conductivity and average heart rate exhibited the greatest decrease. The highest heart rate response rate indicated a mood state change of the participant. Combining physiological data with participants’ psychological preference choices for different landscape types revealed that participants exhibited a large change in N1.1. Compared with the results of the questionnaire, participants exhibited a higher preference for N1.1, indicating that the participants preferred the N1.1 landscape type scene space. Participants had the second highest heart rate response rate in N1.3, and their psychological preference choice score was lowest, indicating that subjects exhibited stronger resistance to N1.3.

Participants exhibited differences in their physiological and psychological responses to different space greening rates. Taken together with another research result of this project, “The impact of spatial canopy closure on the subjects’ physiology” [41]. The current results suggested that, in a real environment, when weather conditions, light conditions, and temperature conditions remain unchanged, spatial canopy closure has a greater impact on users’ physical health. In addition, users experienced the strongest pressure-relieving effect in semi-open and semi-private space. In addition, we found that participants exhibited psychological preferences for open lawn space, and preferred green spaces with some shade and a large field of view.

#### 3.2.2. Analysis of Physiological Indicators with Different Plant Color Scenes

Plant color (N2) was examined using three scene photographs: colorful flower cluster (N2.1), brightly colored ground cover plants (N2.2), and monotonous bushes (N2.3). In the scene photographs, the vividness and richness differed between different plants. We used different plant color scene photographs to examine the effects of plant color on participants’ emotional recovery. The specific scenarios are shown in Table 9.

When participants viewed N2.1, the physiological response rate of skin conductivity, average heart rate, and lowest heart rate exhibited the greatest decrease, and heart rate exhibited the greatest fluctuation. In addition, in the preference questionnaire, participants generally exhibited a stronger preference for N2.1, indicating that bodily stress relief was greatest when viewing colorful flower clusters.

#### 3.2.3. Analysis of Physiological Indicators under Different Landscape Waterscapes

Landscape waterscape (N4) was presented in five scene photographs: the ecological waterscape space with rich plant levels (N4.1), the wooden plank road waterscape space with rich plant levels (N4.2), the ecological waterscape space dotted with classical stone bridges (N4.3), the ecological waterscape space dotted with modern fountains (N4.4), and waterscape space with rigid banks (N4.5). This experiment used different water scene photographs to examine the influence of different ecological and landscape characteristics of water scenes on participants’ emotional state. The specific scenarios are shown in Table 10.

When participants viewed N4.1, the physiological response rate of skin conductivity, average heart rate, and lowest heart rate exhibited the greatest decreases. Taken together with the questionnaire result that participants exhibited the strongest preference for N4.1, these findings indicate that abundant natural waterfront space had the strongest effect on pressure relief.

#### 3.2.4. Analysis of Physiological Indicators in Different Fitness Facility Scenarios

Community park fitness facilities (E1) were presented using three scene photographs: plastic paving material fitness facility activity space (E1.1), stone paving material fitness facility activity space (E1.2) and cement paving material fitness facility activity space (E1.3), to examine the influence of different floor coverings on participants’ emotional state. The specific scenarios are shown in Table 11.

When participants viewed E1.1, the physiological response rate of skin conductivity, average heart rate, and lowest heart rate exhibited the greatest decrease. The results of the questionnaire analysis indicated that participants preferred the equipment fitness space with a plastic floor because of the cushioning effect of the plastic pavement, making participants feel safe. The physiological results revealed that participants exhibited the highest heart rate response rate when watching E1.3. However, the questionnaire revealed that participants had the lowest preference for E1.3, suggesting that the fitness activity space with hard floor equipment may have made participants feel nervous.

#### 3.2.5. Analysis of Physiological Status in Different Fitness Trail Scenarios

The community park fitness trail (E2) was presented in five scene photographs: a plastic paving material fitness trail (E2.1), a walking trail made of floor tiles (E2.2), a semi-cement and semi-plastic dual-material fitness trail (E2.3), hard-paved fitness trails (E2.4), and a stone-paved fitness trail (E2.5), to study the influence of fitness trails of different widths and paving materials on participants’ mood. The specific scenarios are shown in Table 12.

When participants viewed E2.1, the physiological response rate of skin conductivity, average heart rate, and lowest heart rate exhibited the greatest decrease. Combined with the analysis of the preference questionnaire, the results revealed that participants exhibited the greatest degree of pressure relief in relation to the fitness trail with a plastic floor. Plastic pavement has a buffering effect, which can protect participants during exercise and make them feel safe. Participants exhibited the lowest rate of physiological change in response to the bumpy gravel trail, indicating that this trail type had a relatively small stress-relieving effect. In addition, participants’ psychological preference choices for different landscape types revealed that participants exhibited the lowest preference for E1.3, and generally did not prefer fitness activity spaces with hard floors.

#### 3.2.6. Analysis of Physiological Status in Different Activity Space Scenes

Community park activity space (E4) choose four scene photographs, namely a large and open activity space (E4.1), a medium and semi-open activity space (E4.2), a small and private activity space (E4.3), a medium-sized activity space divided by a tree pool (E4.4). The experiment uses different scene photographs of the activity space to study the influence of activity spaces with different areas and site characteristics on the emotions of the participants. The specific scenarios are shown in Table 13.

When participants viewed E4.1, the physiological response rate of skin conductivity and average heart rate exhibited the greatest decrease, indicating that participants felt the most relaxed when viewing a relatively large area of the venue. Participants’ physiological change rate when viewing E4.2 was the second-greatest, indicating that this space also had an effect of relieving participants’ feelings of pressure. Participants exhibited the smallest change in physiological response rate when viewing E4.4, and the average heart rate was higher than that of the control group, indicating that the participants felt nervous when viewing the medium-sized activity space with obstacles. The questionnaire survey revealed that the participants’ spatial preferences were in the following order: large-area venues, medium-area venues, small-area venues, and medium-area venues with obstacles. The results were consistent with the results of the physiological experiment.

#### 3.2.7. Analysis of Physiological Status in Response to Different Resting Seating Facilities

The community park rest seating facility (R1) was presented in five scene photographs, which showed rest seats in a small space beside the road (R1.1), benches with backrests (R1.2), benches without backrests (R1.3), a roadside rest seat (R1.4) and a pavilion rest seat (R1.5), to study the influence of different types of seats and different spatial positions on participants’ emotional state. The specific scenarios are shown in Table 14.

When participants viewed R1.1 stimuli, the physiological response rate of skin conductivity, average heart rate and lowest heart rate exhibited the greatest decreases, indicating that participants felt the least pressure value when viewing the seat under the shade of the tree on the roadside. Participants’ physiological change rate in response to R1.2 was the second-greatest, indicating that viewing the bench with a backrest also relieved participants’ stress. Participants’ physiological response rate to R1.3 exhibited the smallest change, indicating that the bench with no backrest had a relatively small pressure-relieving effect. The results of the preference questionnaire were in accord with those of the physiological test.

#### 3.2.8. Analysis of Physiological Status in Different Pavilions and Corridors

Community park corridor environment (R2) was shown in four scene photographs: modern pavilion (R2.1), new Chinese style corridor (R2.2), Chinese pavilion (R2.3) and new Chinese pavilion (R2.4), to examine the influence of different styles of pavilions on participants’ emotional state. The specific scenarios are shown in Table 15.

When participants viewed R2.3, the physiological response rate of skin conductivity, average heart rate and lowest heart rate exhibited the greatest decreases, indicating that participants felt the least pressure in response to the classical Chinese pavilion space. The second-greatest decrease in the physiological response rate of participants was observed in response to R2.2 stimuli, indicating that the participants experienced stronger relaxation effects in the modern corridor space. The survey results of the preference questionnaire revealed that participants had the strongest preference for the corridor space, followed by the classical Chinese pavilion space. The questionnaire results were largely consistent with the physiological test results.

#### 3.2.9. Analysis of Physiological Status in Different Green Corridor Scenes

The green corridor rest facility (R3) of the community park was presented in three scene photographs: the green corridor full of plants (R3.1), the glass-topped green corridor (R3.2) and the undecorated green corridor (R3.3). We examined the influence of different green corridors on participants’ emotional state. The specific scenarios are shown in Table 16.

When participants viewed R3.1, the physiological response rate of skin conductivity and average heart rate exhibited the greatest decrease, indicating that the participants felt the most relaxed in the green corridor full of plants. Participants’ physiological response rate change value in response to R3.2 was the lowest, indicating that the glass-top green corridor had an unsatisfactory effect on participants’ physiological relaxation. The questionnaire results suggested that participants’ preferences were as follows: green corridor space full of plants, green corridor space without plants, and glass-covered green corridor. These results were in accord with the physiological results.

## 4. Conclusions

Using a psychological questionnaire survey of users, in the current study we constructed a structural equation model of users’ health perceptions regarding the community park landscape environment. The results revealed that the natural environment, activity environment and rest environment in community parks had positive effects on users’ physical health, mental health and social health. At the same time, by examining participants in physiological experiments, we identified types of landscape elements that are helpful for alleviating feelings of pressure among users. Flower clusters and waterscapes in the natural environment landscape, plastic fitness trails and fitness equipment in the sports area, the landscape elements of the rest space such as the benches with backrests, Chinese style pavilions and the green corridor with plants all played positive roles in alleviating feelings of pressure and promoting relaxation among community park users. Based on the current findings, below we propose specific design strategies to improve the health benefits of community park landscapes.

### 4.1. Contributions

The current results revealed that some features of landscape elements helped relieve users’ feelings of pressure and promoted their physical and mental health. These findings are important for informing the design of community parks that promote users’ wellbeing.

In plant landscaping of community parks, the current findings support the use of layered and colorful flower clusters, or bright and uniform ground cover plants. Large areas of brightly colored plants appear to have positive visual effects, in accord with the preferences of community park users. Regarding waterscape design, it is recommended that ecological pools or water surfaces are used to create waterscape spaces. In addition, it is recommended that natural elements, such as wood and stones, are used for the waterfront to avoid ecological interruption and to reduce the artificiality of the waterfront space, forming a unified and coordinated natural waterfront space effect that is not only conducive to the restoration of the ecological environment, but also to the restoration of the physical and mental health of the user.

In the design of rest space, the design of semi-open and semi-private space should be considered. The layout of seats and pavilions in this space should meet the principles of prospect-refuge theory, combined with edge layout and placement, and effectively meet the psychological needs of users for safety and viewing, and reduce anxiety and tension.

Regarding the design of roads in the parks we examined, the main park road is typically used as a fitness trail space, and is constructed from pavement using two materials (semi-plastic and semi-rigid), which can distinguish different sports groups and reduce physical collisions during activities. However, plastic paving is typically used by runners, whereas hard paving is mainly used by older people for walking, which is not conducive to the health of older peoples’ knees. To improve the safety of the park, we recommend the use of two-color plastic materials for paving fitness trails, to distinguish different users’ spaces. In the design of the fitness space, we recommend that the floor of the space be mainly constructed of plastic material, maintaining a certain distance between the activity equipment to reduce interference between activities. It is necessary to provide a variety of fitness settings to attract people to use the park, and to plant natural landscapes such as shrubs and grasses around the equipment fitness activity space to improve the naturalness and safety of the space.

### 4.2. Limitations and Future Research

Innovative aspects of the current study:

The current study took users’ health needs as a starting point and combined measurements of users’ psychological feelings and physiological reactions to examine the health needs of the population at different levels. In addition, we considered the interactive mechanisms of landscape elements and population health comprehensively. The current findings are significant for constructing a theoretical model of healthy landscapes in relation to community parks.

Based on previous physiological experimental research, the current study investigated physiological health recovery in relation to specific types of landscape elements, and clarified the role of specific landscape elements in promoting users’ health recovery.

Although the current study revealed useful findings, it would be valuable for future studies to examine the following issues in more depth.

Expansion of the sample of community parks: The current study selected four community parks in Jing’an District, Shanghai. The selection of research objects could be expanded and improved in future studies. As an old city, Jing’an District’s park model is characteristic of Shanghai’s park development over the last several decades, and the overall layout and model of the city’s community parks tends to be traditional. Constrained by the status quo of the region and the degree of urbanization, the community parks in Jing’an District can only be renovated, rather than newly built. In recent years, with the expansion of cities, the development model of community parks in high-density urban high-tech zones is also of great research interest.

Improvement of the physiological test experiment: Unlike many physiological studies, the current physiological experiment did not involve a stressor. With longer test times, even if there was no relaxing effect of the landscape, the stress felt by participants would be expected to decrease over time. Participants in this experiment only spent 10 s watching the scene photographs. Therefore, we used physiological measures of skin conductivity and heart rate, which can instantly reflect the current physiological state of participants as the test indicators. In future experiments, experimenters should consider using eye trackers, electroencephalography, and other equipment for testing, which can more fully reflect the physiological conditions of the participants.

Improving the authenticity of scene simulation: In this study, a photograph viewing method was used to examine the health benefits of landscape elements. However, this simulation method may have lacked authenticity in some respects. Community parks are places containing diverse types of sensory information. Future research should examine all five sensory domains: vision, hearing, smell, touch, and taste. Further studies should test participants in real environments, or using VR and other approaches to simulate more realistic scenes.

## Figures and Tables

**Figure 1 ijerph-18-07888-f001:**
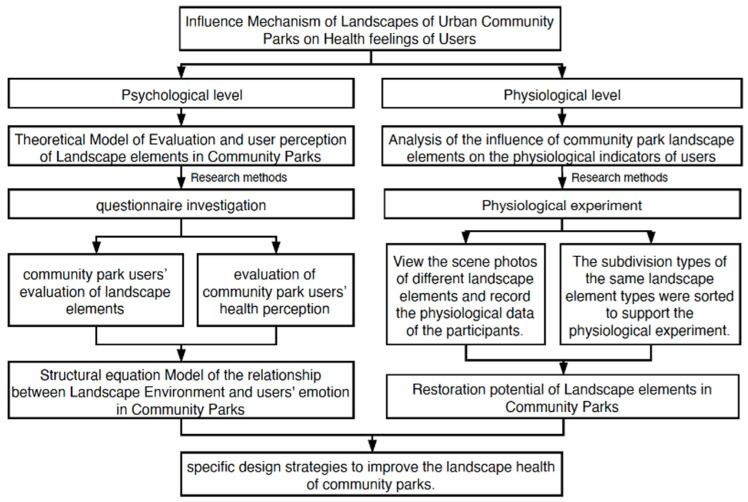
Schematic diagram of research methods to investigate the mechanisms underlying the interactions between the environment and health of community park users.

**Figure 2 ijerph-18-07888-f002:**
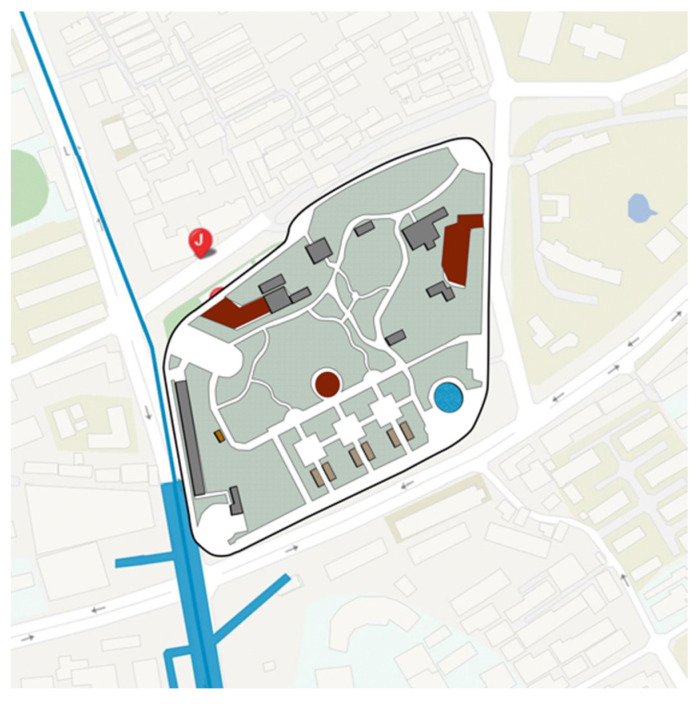
Map of Zhongxin Greenland.

**Figure 3 ijerph-18-07888-f003:**
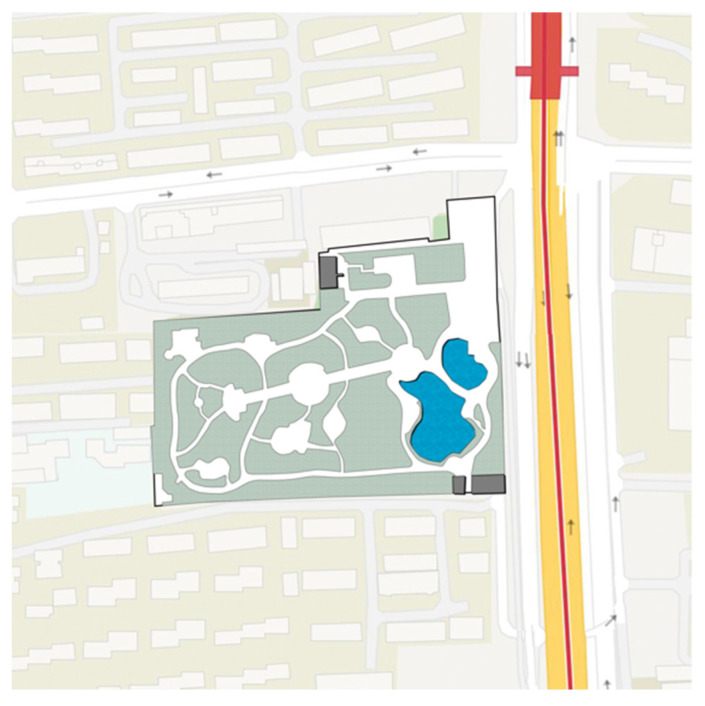
Map of Pengpu Park.

**Figure 4 ijerph-18-07888-f004:**
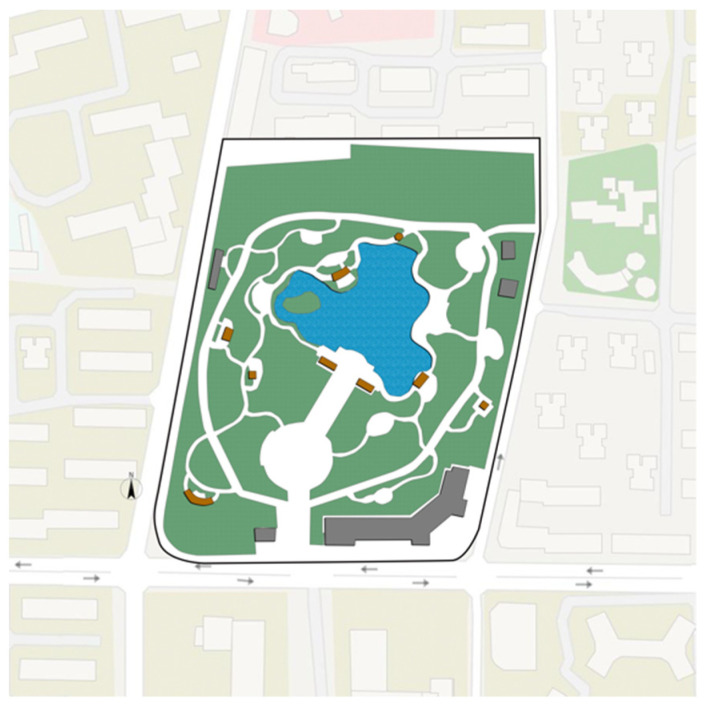
Map of Lingnan Park.

**Figure 5 ijerph-18-07888-f005:**
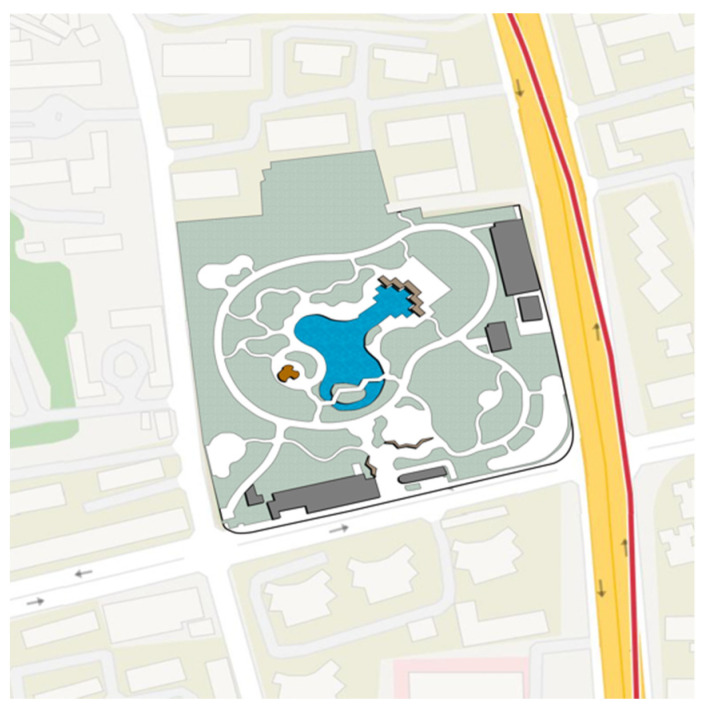
Map of Sanquan Park.

**Figure 6 ijerph-18-07888-f006:**
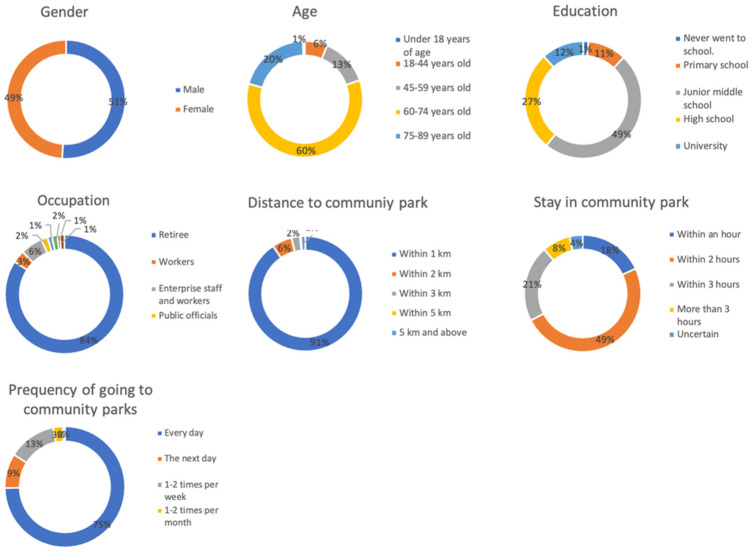
Basic information of community park users.

**Figure 7 ijerph-18-07888-f007:**
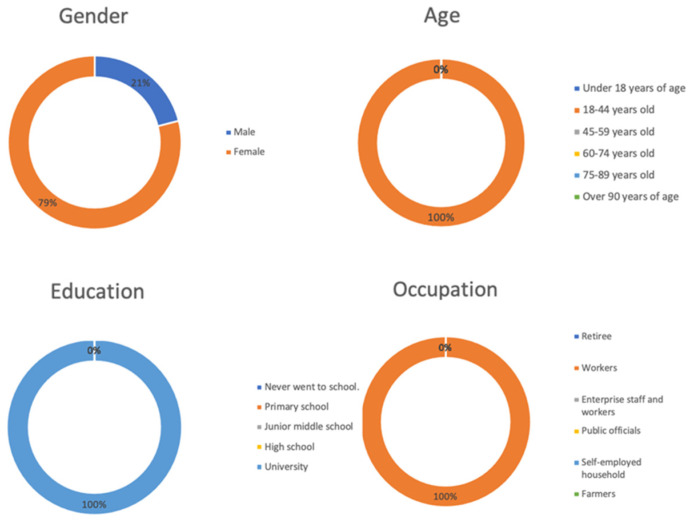
Basic information of participants in the physiological experiment.

**Figure 8 ijerph-18-07888-f008:**
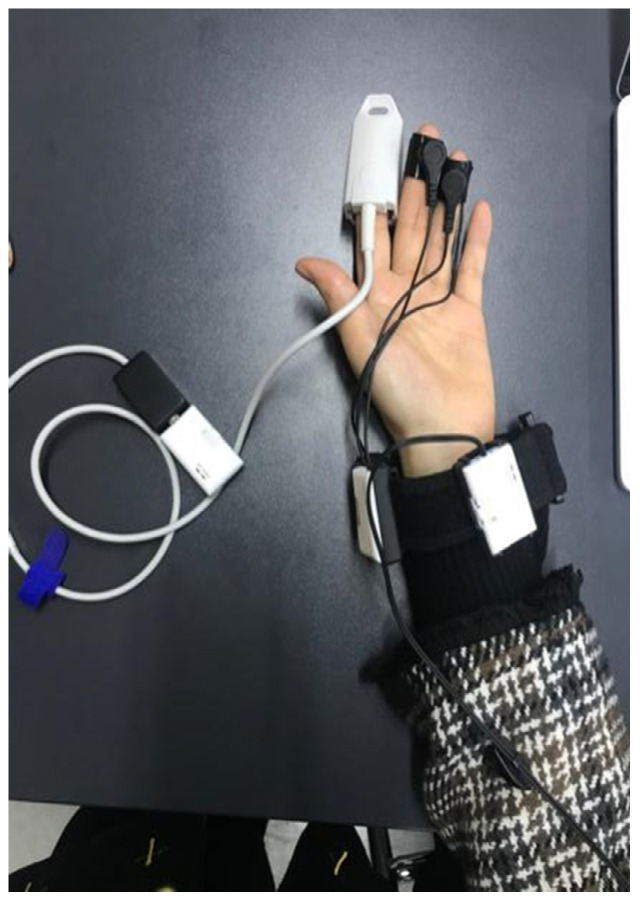
The subject wore wireless physiological sensors.

**Figure 9 ijerph-18-07888-f009:**
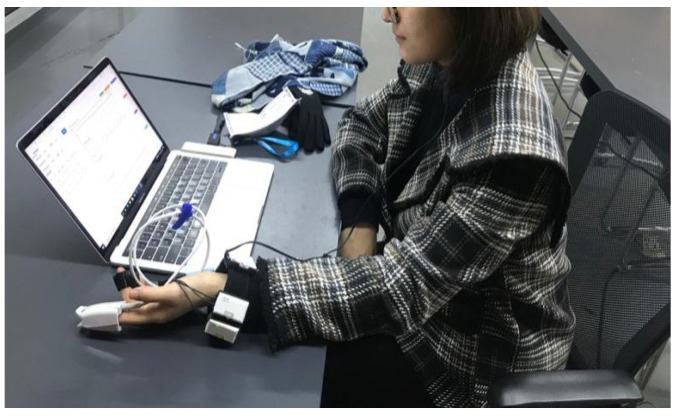
The participant sat still and waited for fluctuations of the physiological experimental data to stabilize.

**Figure 10 ijerph-18-07888-f010:**
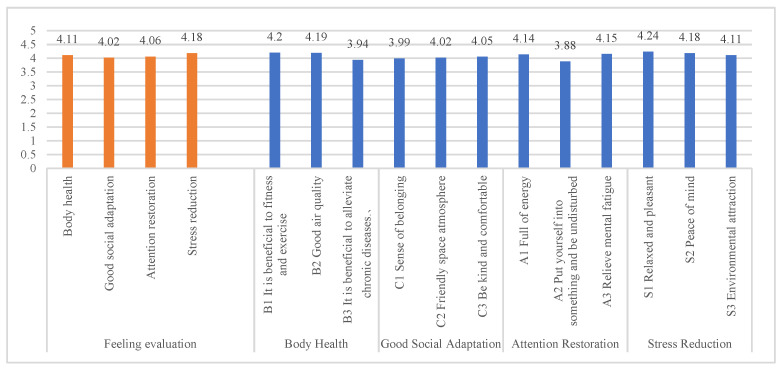
Results of user perception evaluation in community parks.

**Figure 11 ijerph-18-07888-f011:**
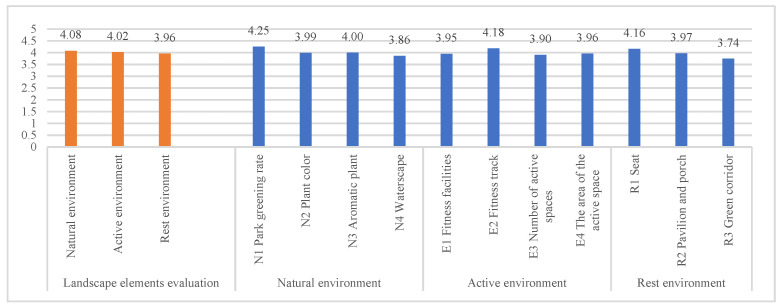
Evaluation statistics of landscape elements by community park users.

**Figure 12 ijerph-18-07888-f012:**
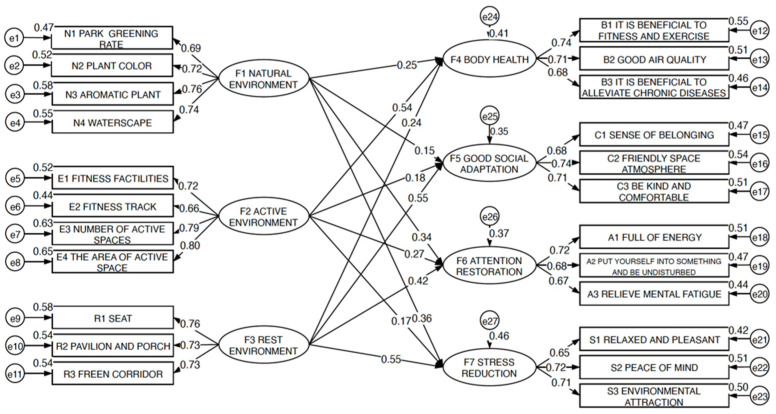
Standardized parameter estimation path map of community park landscape elements evaluation and user perception theory model.

**Table 1 ijerph-18-07888-t001:** Classification of landscape elements in community parks.

Community Park	Landscape Space	Landscape Elements
The overall environment of the park	Natural environment	N1 Park greening rate
		N2 Plant color
		N3 Aromatic plant
		N4 Waterscape
	Active environment	E1 Fitness facilities
		E2 Fitness track
		E3 Number of active spaces
		E4 The area of the active space
	Rest environment	R1 Seat
		R2 Pavilion and porch
		R3 Green corridor

**Table 2 ijerph-18-07888-t002:** Evaluation indicators of feelings of community park users.

Target Layer	Programme Layers		Criterion Layers
Community park user perception indicators	Body Health		B1 It is beneficial to fitness and exercise.
			B2 Good air quality
			B3 It is beneficial to alleviate chronic diseases
	Good Social Adaptation		C1 Sense of belonging
			C2 Friendly space atmosphere
			C3 Be kind and comfortable
	Mental health	Attention Restoration	A1 Full of energy
			A2 Put yourself into something and be undisturbed
			A3 Relieve mental fatigue
		Stress Reduction	S1 Relaxed and pleasant
			S2 Peace of mind
			S3 Environmental attraction

**Table 3 ijerph-18-07888-t003:** Experimental flow.

Experimental Steps	Time	Participants’ Activities
Experimental preparation phase	5 min	Listen to the experiment instructions, fill in the experiment consent form, personal information form, and wear physiological experiment equipment
	3 min	Sitting still and waiting for the fluctuation of physiological experiment data to stabilize
Experimental test stage	10 min	View photos of landscape elements
	2 min	Take off the physiological experiment equipment for the participants
Questionnaire test stage	5 min	Fill out the test feeling questionnaire

**Table 4 ijerph-18-07888-t004:** Physiological measurement indicators and response mechanisms.

Physiological Indicators	Physiological Response Mechanism
Skin conductivity	Skin conductivity represents the resistance value or the amount of current passing between two points of the human skin, which can reflect the changes in human emotions [38]. When the human body is in a state of tension or emotional anxiety, the human body’s sympathetic nerves are in a state of excitement, which leads to the activity of sweat glands increases, the secretion of sweat increases, and the value of skin electricity increases [39].
Heart rate	The number of times the human heart beats per unit time is called heart rate. The heart rate of the human body can reflect the degree of contraction of the human cardiovascular system. When the human body is in a state of emotional tension, anxiety or excitement, the human heart rate increases [40].

**Table 5 ijerph-18-07888-t005:** Reliability test for questionnaire scale.

Measured Variable	Measured Indicators	Cronbach α Coefficient
Body Health	B1~B3	0.788
Good Social Adaptation	C1~C3	0.777
Attention Restoration	A1~A3	0.768
Stress Reduction	S1~S3	0.769
Natural Environment	N1~N4	0.822
Active Environment	E1~E4	0.833
Rest Environment	R1~R3	0.804

**Table 6 ijerph-18-07888-t006:** KMO and Bartlett’s test.

Take the Kaiser–Meyer–Olkin metric of sufficient degree		0.913
Bartlett’s sphericity test	Approximate Chi-square	4479.351
	Df	276
	Sig	0.000

**Table 7 ijerph-18-07888-t007:** Analysis of interpretation ability of latent variable indicators.

Measurement Indicators	Estimate	*p*
Variances	positive numbers	*** (Significant)
Regression Weights	positive numbers	<0.05
Factor loading	0.65~0.80	––

**Table 8 ijerph-18-07888-t008:** Physiological indicators of participants before and after the measurement of greening rate in different groups of parks.

Measurement Indicators	Types of Landscape Elements (Experimental Group)			Control Group
N1	N1.1 ** 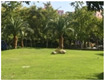 **	N1.2 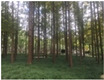	N1.3 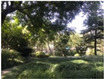	Baseline control
Preference score	2.52	1.9	1.54	––
SC/μS	2.38 ± 2.20 **	2.32 ± 2.16 **	2.16 ± 1.99 **	2.23 ± 2.07
R	6.73%	4.04%	−3.14%	——
HR/bpm	85.90 ± 17.35 **	83.33 ± 14.27 *	82.53 ± 13.22 **	85.22 ± 10.59
HRmax/bpm	90.41 ± 18.75**	84.11 ± 17.10 **	86.00 ± 16.81 **	86.43 ± 19.41
HRmin/bpm	70.46 ± 8.07 **	70.20 ± 7.31 **	71.69 ± 6.99 **	73.31 ± 7.36

The data in the table are shown as mean ± standard deviation. * Indicates significance at a level of *p* < 0.05. ** Indicates significance at a level of *p* < 0.01.

**Table 9 ijerph-18-07888-t009:** Physiological indicators of participants before and after the measurement of plant color in different groups.

Measurement Indicators	Types of Landscape Elements (Experimental Group)			Control Group
N2	N2.1 ** 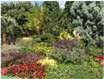 **	N2.2 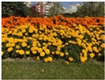	N2.3 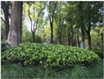	Baseline control
Preference score	2.15	2.04	1.73	––
SC/μS	2.11 ± 1.93 **	2.11 ± 2.00 **	2.13 ± 2.22 **	2.23 ± 2.07
R	−5.38%	−5.38%	−4.48%	––
HR/bpm	83.20 ± 12.43 *	84.02 ± 12.95 *	84.47 ± 16.17 *	85.22 ± 10.59
HRmax/bpm	88.59 ± 15.05 **	86.27 ± 17.94 *	84.84 ± 15.18 **	86.43 ± 19.41
HRmin/bpm	70.35 ± 8.14 *	71.98 ± 7.25 **	71.80 ± 7.75 *	73.31 ± 7.36

The data in the table are shown as the mean ± standard deviation. * Indicates significance at a level of *p* < 0.05. ** Indicates significance at a level of *p* < 0.01.

**Table 10 ijerph-18-07888-t010:** Physiological indicators of participants before and after the measurement in response to different waterscape spaces.

Measurement Indicators	Types of Landscape Elements (Experimental Group)					Control Group
N4	N4.1 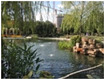	N4.2 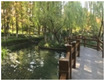	N4.3 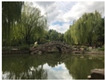	N4.4 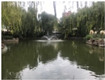	N4.5 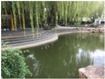	Baseline control
Preference score	3.52	3.27	3.44	2.08	2.67	––
SC/μS	2.04 ± 2.20 **	2.08 ± 2.30 **	2.04 ± 2.23 **	2.10 ± 2.31 **	2.11 ± 2.22 **	2.23 ± 2.07
R	−8.52%	−6.73%	−8.52%	−5.83%	−5.38%	––
HR/bpm	82.02 ± 13.91 **	83.77 ± 16.58 **	83.28 ± 14.25 **	84.59 ± 16.09 **	84.13 ± 14.97 **	85.22 ± 10.59
HRmax/bpm	87.76 ± 15.27 *	88.24 ± 17.29 *	85.24 ± 15.44 **	86.92 ± 17.80 **	88.24 ± 17.37 **	86.43 ± 19.41
Hrmin/bpm	71.69 ± 8.74 *	71.80 ± 7.60 **	72.87 ± 8.50 **	73.65 ± 7.65 **	72.76 ± 7.31 *	73.31 ± 7.36

The data in the table are shown as the mean ± standard deviation. * Indicates significance at a level of *p* < 0.05. ** Indicates significance at a level of *p* < 0.01.

**Table 11 ijerph-18-07888-t011:** Physiological indicators of participants before and after the measurement in response to different fitness facility spaces.

Measurement Indicators	Types of Landscape Elements (Experimental Group)			Control Group
E1	E1.1 ** 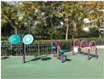 **	E1.2 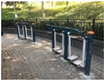	E1.3 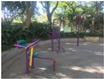	Baseline control
Preference score	2.04	2.10	1.85	––
SC/μS	1.97 ± 2.14 **	2.03 ± 2.23 **	1.98 ± 2.09 **	2.23 ± 2.07
R	−11.66%	−8.97%	−11.21%	––
HR/bpm	83.02 ± 20.19 **	83.41 ± 13.01 **	83.68 ± 12.35 **	85.22 ±10.59
HRmax/bpm	85.35 ± 16.98 **	85.86 ± 16.77 **	89.73 ± 17.87 **	86.43 ± 19.41
HRmin/bpm	72.26 ± 7.65 **	72.87 ± 7.49 **	72.91 ± 7.64 **	73.31 ± 7.36

The data in the table are shown as the mean ± standard deviation. * Indicates significance at a level of *p* < 0.05. ** Indicates significance at a level of *p* < 0.01.

**Table 12 ijerph-18-07888-t012:** Physiological indicators of participants before and after the measurement in response to different fitness trail spaces.

Measurement Indicators	Types of Landscape Elements (Experimental Group)					Control Group
E2	E2.1 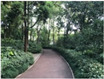	E2.2 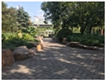	E2.3 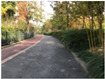	E2.4 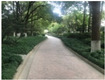	E2.5 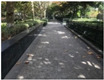	Baseline control
Preference score	3.71	3.04	3.08	3.04	2.08	––
SC/μS	2.03 ± 2.27 **	2.06 ± 2.34 **	2.05 ± 2.32 **	2.06 ± 2.42 **	2.08 ± 2.50 **	2.23 ± 2.07
R	−8.97%	−7.62%	−8.07%	−7.62%	−6.73%	––
HR/bpm	80.97 ± 10.96 *	83.41 ± 13.68 **	81.96 ± 12.79 **	82.55 ± 12.79 **	84.35 ± 12.14 **	85.22 ±10.59
HRmax/bpm	87.08 ± 15.94 **	83.78 ± 14.27 **	89.81 ± 16.76 *	86.19 ± 16.35 **	85.65 ± 17.50 **	86.43 ± 19.41
HRmin/bpm	71.35 ± 8.05 **	72.31 ± 7.42 **	71.31 ± 7.04 **	72.30 ± 7.30 **	72.80 ± 8.20 **	73.31 ± 7.36

The data in the table are shown as the mean ± standard deviation. * Indicates significance at a level of *p* < 0.05. ** Indicates significance at a level of *p* < 0.01.

**Table 13 ijerph-18-07888-t013:** Physiological indicators of participants before and after the measurement in response to different activity spaces.

Measurement Indicators	Types of Landscape Elements (Experimental Group)				Control Group
E4	E4.1 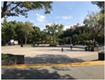	E4.2 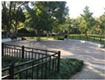	E4.3 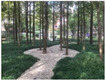	E4.4 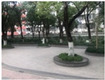	Baseline control
Preference score	3.21	2.75	2.1	1.94	––
SC/μS	2.06 ± 2.43 **	2.11 ± 2.54 **	2.12 ± 2.68 **	2.13 ± 2.71 **	2.23 ± 2.07
R	−7.62%	−5.38%	−4.93%	−4.48%	––
HR/bpm	83.13 ± 15.29 **	84.37 ± 16.26 **	85.39 ± 16.29 **	86.31 ± 17.76 *	85.22 ± 10.59
HRmax/bpm	88.43 ± 16.67 *	86.95 ± 16.73 **	85.38 ± 17.11 **	87.11 ± 17.60 **	86.43 ± 19.41
HRmin/bpm	72.26 ± 7.74 **	72.87 ± 6.35 **	72.26 ± 7.17 **	71.09 ± 7.90 **	73.31 ± 7.36

The data in the table are shown as mean ± standard deviation. * Indicates significance at a level of *p* < 0.05. ** Indicates significance at a level of *p* < 0.01.

**Table 14 ijerph-18-07888-t014:** Physiological indicators of participants before and after measurement in response to different landscape seating spaces.

Measurement Indicators	Types of Landscape Elements (Experimental Group)					Control Group
R1	R1.1 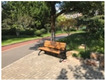	R1.2 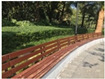	R1.3 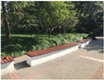	R1.4 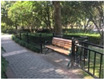	R1.5 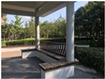	Baseline control
Preference score	3.29	2.9	2.44	2.83	3.52	––
SC/μS	2.06 ± 2.44 **	2.08 ± 2.38 **	2.10 ± 2.41 **	2.10 ± 2.41 **	2.18 ± 2.48 **	2.23 ± 2.07
R	−7.62%	−6.73%	−5.83%	−5.83%	−2.24%	––
HR/bpm	81.84 ± 12.15 *	83.13 ± 15.29 **	83.40 ± 13.63 **	83.52 ± 12.99 **	83.12 ± 14.24 *	85.22 ± 10.59
HRmax/bpm	85.38 ± 16.86 **	84.32 ± 14.98 **	86.22 ± 16.43 **	85.46 ± 15.80 **	86.16 ± 15.22 **	86.43 ± 19.41
HRmin/bpm	71.96 ± 7.26 *	73.11 ± 8.40 **	72.69 ± 8.69 *	73.31 ± 8.09 **	72.54 ± 7.43 **	73.31 ± 7.36

The data in the table are shown as mean ± standard deviation. * Indicates significance at a level of *p* < 0.05. ** Indicates significance at a level of *p* < 0.01.

**Table 15 ijerph-18-07888-t015:** Physiological indicators of participants before and after the measurement in response to different pavilions and corridors.

Measurement Indicators	Types of Landscape Elements (Experimental Group)				Control Group
R2	R2.1 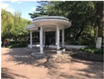	R2.2 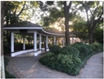	R2.3 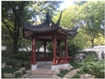	R2.4 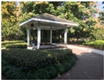	Baseline control
Preference score	2.5	2.69	2.63	2.17	––
SC/μS	2.22 ± 2.58 **	2.22 ± 2.62 **	2.21 ± 2.55 **	2.34 ± 2.92 **	2.23 ± 2.07
R	−0.45%	−0.45%	−0.90%	4.93%	––
HR/bpm	86.68 ± 18.33 *	84.22 ± 17.32 *	81.56 ± 13.42	86.26 ± 13.82 **	85.22 ± 10.59
HRmax/bpm	84.05 ± 15.02 **	87.70 ± 17.53 **	90.00 ± 17.89 **	86.65 ± 17.92 **	86.43 ± 19.41
HRmin/bpm	72.24 ± 7.51 *	72.87 ± 7.94 **	71.35 ± 7.48 **	72.52 ± 8.84 **	73.31 ± 7.36

The data in the table are shown as mean ± standard deviation. * Indicates significance at a level of *p* < 0.05. ** Indicates significance at a level of *p* < 0.01.

**Table 16 ijerph-18-07888-t016:** Physiological indicators of participants before and after measurement in response to different green corridor spaces.

Measurement Indicators	Types of Landscape Elements (Experimental Group)			Control Group
R3	R3.1 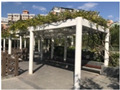	R3.2 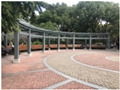	R3.3 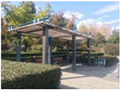	Baseline control
Preference score	2.31	1.79	1.88	––
SC/μS	2.27 ± 2.78 **	2.33 ± 2.84 **	2.28 ± 2.73 **	2.23 ± 2.07
R	1.79%	4.48%	2.24%	––
HR/bpm	85.41 ± 15.76 **	85.82 ± 14.07 *	85.51 ± 15.69 **	85.22 ± 10.59
HRmax/bpm	85.24 ± 15.07	86.73 ± 17.97 **	84.43 ± 16.30	86.43 ± 19.41
HRmin/bpm	73.20 ± 7.58 **	74.46 ± 8.81 **	73.44 ± 7.35 **	73.31 ± 7.36

The data in the table are presented as mean ± standard deviation. * Indicates significance at a level of *p* < 0.05. ** Indicates significance at a level of *p* < 0.01.

## Data Availability

The data presented in this study are available on request from the corresponding author. As physiological data involves the privacy of participants, the data has not been made publicly available.

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
