# Peer review of "Mechanisms Underlying the Effects of Landscape Features of Urban Community Parks on Health-Related Feelings of Users"

_ijerph, 2021, doi:10.3390/ijerph18157888_

Round 1

Reviewer 1 Report

The paper has applied an interesting method to understand preferences. Comments and considerations below:

  • The paper starts with a good review of related literature. However, it does not highlight the gap in the literature and why you continued the new inquiry through your research questions. Good and necessary to emphasis on this.
  • The analysis findings (section 3.2.1) are interesting (differences between physio. & psych.) is there any previous research finding to compare to (ie. In relation to canopy cover). Also what is the climatic condition, is sun more preferred than shaded area? Good to relate the climatic conditions. In subtropical environment, canopy cover can be a positive factor compared to exposed sun area (what can be positive for respondents?)
  • What questions have been asked in the survey form? What the preference range? 1 to 5 (lowest to highest)?
  • Needs a summary of findings, where physio. VS psych. are conforming or not.
  • In conclusion, were physio. measurements useful/helpful/conforming for further research?
  • highlight key contributions of the research. what is the innovation here?

Author Response

1、The paper starts with a good review of related literature. However, it does not highlight the gap in the literature and why you continued the new inquiry through your research questions. Good and necessary to emphasis on this.

The current literature research is from rough to detailed. At present, this paper is more detailed on the basis of the original, to explore the effect of the same type of landscape subdivision type on the user's pressure relief.such as different colors of flowers, different combination patterns of waterscape spaces, different paving materials, and different areas of playgrounds,to clarify the impact of different types of landscape elements on the health of participants.

2、The analysis findings (section 3.2.1) are interesting (differences between physio. & psych.) is there any previous research finding to compare to (ie. In relation to canopy cover). Also what is the climatic condition, is sun more preferred than shaded area? Good to relate the climatic conditions. In subtropical environment, canopy cover can be a positive factor compared to exposed sun area (what can be positive for respondents?)

The physiological and psychological responses of the participants were different under different greening rates. Combined with another research achievement of this project "the effect of spatial canopy closure on the physiology of subjects" [41], the experiment proved that in the real environment, when the weather conditions, light conditions and temperature conditions remain unchanged, the spatial canopy density has a great impact on the physiological health of users. Users in semi open and semi private space have the best pressure relief effect. Combined with this experiment, it was found that participants preferred open lawn space psychologically and green space with certain shade and vision physiologically.

3、What questions have been asked in the survey form? What the preference range? 1 to 5 (lowest to highest)?

The grading design of the questionnaire items adopts the Likert five-level scale. The questionnaire items focus on three aspects: basic information of community park users, evaluation of community park users’ health perception, and community park users’ evaluation of landscape elements. Regarding the health perception evaluation of community park users, the questionnaire surveyed the health status of users after their activities in the park. Specific questions are shown in Table 2. In terms of community park users’ evaluation of landscape elements, a questionnaire surveyed users’ perceptions and evaluations of different landscape elements. The specific types of landscape elements are shown in Table 1. Among them, the items are divided into five levels, namely "very agree", "very agree", "general", "disagree", and "hate disagree". The corresponding points are 5 to 1 points.

4、Needs a summary of findings, where physio. VS psych. are conforming or not.

In the physiological experiment research, the physiological test and physiological test in the green vision test experiment are not exactly the same. In other experiments, the results of physiological and psychological tests were almost the same. This part will be added to the paper.

5、In conclusion, were physio measurements useful/helpful/conforming for further research?

The F test in SPSS analysis showed that there were significant differences in skin conductivity, average heart rate, maximum heart rate and minimum heart rate.Therefore, by measuring the physiological data of the participants, this experiment can obtain the pressure relief effect of the participants under the scene photos of different landscape elements.

6、highlight key contributions of the research. what is the innovation here?

The innovations of this article:

This article takes the user's health needs as the starting point, combines the user's psychological feelings and physiological reactions, studies the health needs of the population at different levels, discusses the interactive mechanism of landscape elements and population health in all aspects, which has positive significance for constructing a theoretical model of a community park's healthy landscape.

On the basis of the existing physiological experimental research, this paper further studies the physiological health recovery of the participants by the specific types of landscape elements, and clarifies the role of specific landscape elements in promoting the health recovery of users.

Reviewer 2 Report

I enjoyed reading this manuscript. The paper meets the goals of the International Journal of Environmental Research and Public Health. The topic is of interest to the community that focuses on public space, and the results can also help design better spaces in the future.

On the other hand, I have some comments, mainly in the methodology section, with what I think the authors can raise the quality of the paper:

1. The basic information of the 50 participants needs to be displayed, as was done in Figure 1.

2. It is convenient to add a map with the location of the parks. It would also be convenient to add on each map the location of the photos shown in the experiment.

3. Finally, it would be very useful for the reader to include a diagram showing the methodology followed throughout the experiment.

Author Response

1、The basic information of the 50 participants needs to be displayed, as was done in Figure 1

It will be shown in the article.

In this study, community park users were not invited to conduct a real study.

First, because community park users are mostly elderly people who have certain visual impairments, when viewing scene photos of landscape elements, physiological data may not accurately reflect their feelings during the test.

Second, most elderly people have a certain degree of physical disease, such as high blood pressure, diabetes, heart disease, etc. It is difficult to find a sufficient number of elderly volunteers with complete health.

Based on the above reasons, this article mainly invites college students to conduct physiological experiments. College students can better understand the intention of the experiment, have a high degree of cooperation, and are in good health, making them more suitable for physiological experiments.

Since the basic data of college students are relatively consistent and have no major correlation with physiological experiments, they are not listed.

 2、It is convenient to add a map with the location of the parks. It would also be convenient to add on each map the location of the photos shown in the experiment.

It will be shown in the article.

3、Finally, it would be very useful for the reader to include a diagram showing the methodology followed throughout the experiment.

It will be shown in the article.

Reviewer 3 Report

A very interesting paper with a sophisticated method, well designed and presented. After minor corrections, it is ready to be published.

The introduction Is well structured, informative and interesting. It will be good to provide some definitions: what is a natural landscape? What is an artificial landscape? How to choose elements of these landscapes using their differences? I appreciate the description of the material and methods.  A bit confusing is choose of landscape elements shown in table 1. There are different types of factors with different units (m2, pcs., %), please convince me that you can cope with it. Please also say how it is possible to distribute 440 questionnaires and receive 440? What was the method of providing questionnaires? 

The results are clear and well presented as well. Please add Scenic Beauty Estimation (1976) method to the discussion. There is one confusing part of conclusions: what is the place of territorial domains in that research (lines 513-517)?

A subchapter 2.4.2 (line 180-187) should be after 2.4.3.

Author Response

1、A bit confusing is choose of landscape elements shown in table 1. There are different types of factors with different units (m2, pcs., %), please convince me that you can cope with it.

In the revised paper will use a more clear chart display

2、Please also say how it is possible to distribute 440 questionnaires and receive 440? What was the method of providing questionnaires? 

In the questionnaire survey, because the elderly have certain visual impairment, the experimenter explained the questionnaire for the users, and then filled in the questionnaire. The experimenter asked thousands of users whether they would like to participate in the questionnaire survey, and finally 440 users were willing to participate in the questionnaire survey, so 440 questionnaires were obtained

Reviewer 4 Report

Dear Authors,

The manuscript develops a topic of interest but the following points need to be revised:
1 - The abstract needs to be rewritten in order to be clear about the objective, methodology and findings achieved;
2 - The conclusion as written does not make it understandable what the research ends are;
3 - English must be revised in order to avoid high repetition of expressions such as "it is recommended" which is repeated at least 3 times in the conclusions.

Author Response

1 - The abstract needs to be rewritten in order to be clear about the objective, methodology and findings achieved;

Abstract: Urban community parks are closely related to the healthy life of residents, and have a positive effect on residents' perception of nature, alleviating anxiety, and promoting physical health. At present, there are many literatures on the impact of community parks on the health of the population, but there are few related literature studies on the potential of specific types of landscape elements in community parks to restore the human body. We obtained psychological feeling questionnaires from 440 users in Shanghai community parks through on-site surveys. Based on the psychological feeling questionnaire, a structural equation model of the relationship between the community park landscape environment and users’ feelings was established, proving that the natural environment, activity environment, and rest environment in the community park have a positive impact on the physical, mental, and social health of users . At the same time, we recruited 50 volunteers to conduct laboratory experiments to study the physiological data changes of volunteers when viewing different types of scene photos of the same landscape element. By measuring their physiological indicators, such as skin conductivity and heart rate, we can derive the types of landscape elements that help relieve the stress of users.The results show that: the flower cluster and waterscape in the natural environment landscape, the plastic fitness trails and fitness equipment places in the sports space, the landscape elements of rest space such as the backrest seats, Chinese style pavilions and the green corridor with plants play positive roles in alleviating the pressure and relaxation of community park users. Based on the research conclusions, this article puts forward specific design strategies to improve the landscape health of community parks.

2 - The conclusion as written does not make it understandable what the research ends are;

Through the psychological questionnaire survey of users, this paper constructs the structural equation model of user health perception and community park landscape environment, and proves that the natural environment, activity environment and rest environment in community park have positive effects on users' physical health, mental health and social health. At the same time, through the physiological experiments of the participants, this paper obtains the types of landscape elements that are helpful to alleviate the pressure of the population.The flower cluster and waterscape in the natural environment landscape, the plastic fitness trails and fitness equipment places in the sports space, the landscape elements of rest space such as the backrest seats, Chinese style pavilions and the green corridor with plants play positive roles in alleviating the pressure and relaxation of community park users. Based on the research conclusions, this article puts forward specific design strategies to improve the landscape health of community parks.

3 - English must be revised in order to avoid high repetition of expressions such as "it is recommended" which is repeated at least 3 times in the conclusions.

This English will be modified in the article.

Round 2

Reviewer 4 Report

Dear Authors,

Thank you for the amazing improvment made in your initial text.

The new text calrify my previous questions and suggestions.